# Structural and functional insights into the *bona fide* catalytic state of *Streptococcus pyogenes* Cas9 HNH nuclease domain

Zhicheng Zuo[1,2], Ashwini Zolekar[1], Kesavan Babu[3], Victor JT Lin[1], Hamed S Hayatshahi[1], Rakhi Rajan[3], Yu-Chieh Wang[1]*, Jin Liu[1]*

[1]Department of Pharmaceutical Sciences, UNT System College of Pharmacy, University of North Texas Health Science Center, Fort Worth, United States; [2]College of Chemistry and Chemical Engineering, Shanghai University of Engineering Science, Shanghai, China; [3]Department of Chemistry and Biochemistry, Price Family Foundation Institute of Structural Biology, Stephenson Life Sciences Research Center, University of Oklahoma, Norman, United States

**Abstract** The CRISPR-associated endonuclease Cas9 from *Streptococcus pyogenes* (SpyCas9), along with a programmable single-guide RNA (sgRNA), has been exploited as a significant genome-editing tool. Despite the recent advances in determining the SpyCas9 structures and DNA cleavage mechanism, the cleavage-competent conformation of the catalytic HNH nuclease domain of SpyCas9 remains largely elusive and debatable. By integrating computational and experimental approaches, we unveiled and validated the activated Cas9-sgRNA-DNA ternary complex in which the HNH domain is neatly poised for cleaving the target DNA strand. In this catalysis model, the HNH employs the catalytic triad of D839-H840-N863 for cleavage catalysis, rather than previously implicated D839-H840-D861, D837-D839-H840, or D839-H840-D861-N863. Our study contributes critical information to defining the catalytic conformation of the HNH domain and advances the knowledge about the conformational activation underlying Cas9-mediated DNA cleavage.
DOI: https://doi.org/10.7554/eLife.46500.001

*For correspondence:
yu-chieh.wang@unthsc.edu (Y-CW);
jin.liu@unthsc.edu (JL)

**Competing interests:** The authors declare that no competing interests exist.

## Introduction

The clustered regularly interspaced short palindromic repeats (CRISPR)-associated endonuclease Cas9 from *Streptococcus pyogenes* (SpyCas9) has become a gene-editing tool that holds an immense promise for the development of novel therapeutic approaches for human diseases (*Cong et al., 2013*; *Jinek et al., 2012*; *Knott and Doudna, 2018*; *Mali et al., 2013*). Two, magnesium (Mg[2+]) ion-dependent, nuclease domains (*i.e.* HNH and RuvC domains) in Cas9 cleave the target DNA strand (tDNA) complementary to the guide region of a dual single-guide RNA (sgRNA) and the non-target DNA strand (ntDNA), respectively (*Gasiunas et al., 2012*; *Jinek et al., 2012*). The conformational and mechanistic knowledge of Cas9 activation to achieve DNA cleavage is essential for the rational engineering of Cas9 to possibly ensure minimal off-target effects while retaining high gene-editing efficiency (*Chen et al., 2017a*; *Kleinstiver et al., 2016*; *Slaymaker et al., 2016*; *Sternberg et al., 2015*).

Several structures of the SpyCas9-sgRNA-DNA ternary complex to depict the HNH nuclease domain in a 'cleavage-competent' state have been reported (*Huai et al., 2017*; *Palermo et al., 2018*; *Palermo et al., 2017*; *Zuo and Liu, 2017*). Notably, the amino acid residue D861 in the HNH domain of SpyCas9 pointed towards the catalytic center in the absolute majority of resolved crystal structures (*Anders et al., 2016*; *Anders et al., 2014*; *Dong et al., 2017*; *Hirano et al., 2016*; *Jiang et al., 2015*; *Jiang et al., 2016*; *Jinek et al., 2014*; *Liu et al., 2019*; *Nishimasu et al., 2014*;

**eLife digest** The DNA inside human cells provides instructions for all of the processes that happen inside the body. Errors in the DNA may lead to cancer, sickle cell disease, cystic fibrosis, Huntington's disease, or other genetic disorders. Medical researchers are exploring whether it is possible to replace or repair the faulty DNA (an approach known as gene therapy) to reduce the symptoms, or even cure individuals, of these conditions.

Over the last ten years, a new technology known as CRISPR-Cas9 gene editing has proved to be a reliable and efficient way to make small and precise changes to DNA in living cells. First, an enzyme called Cas9 searches for a segment of target DNA segment that matches a template molecule the enzyme carries. Cas9 then cuts the target DNA, which is repaired to match a new customized DNA sequence: this changes the genetic information of the cell.

The Cas9 protein is made of a succession of building blocks called amino acids that create long chains which then fold to form the final three-dimensional shape of the enzyme. A region of Cas9 known as the HNH domain is responsible for cutting the target DNA. However, it remains unclear exactly which amino acids within this domain work together to sever the DNA.

Here, Zuo et al. combined computational and experimental approaches to reveal the three-dimensional structure of the Cas9 enzyme when the HNH domain is poised to cut the target DNA. The findings were used to generate a computational model of Cas9 and this model predicted that the HNH domain relies on a group of three amino acids known collectively as D839-H840-N863 to cleave DNA strands.

This knowledge is useful to understand exactly how Cas9 modifies genetic information. Ultimately, this may help to improve CRISPR-Cas9 technology so it could be safely used in geneediting therapies.

DOI: https://doi.org/10.7554/eLife.46500.002

---

Olieric et al., 2016; Yang and Patel, 2017), molecular dynamic simulation models (Palermo et al., 2018; Zuo and Liu, 2017), and cryo-electron microscopy (cryo-EM) structures (Huai et al., 2017; Jiang et al., 2019; Shin et al., 2017) (Figure 1a–1d). Despite lacking experimental evidence, it is generally believed that D861 directly participates in $Mg^{2+}$ chelation and tDNA cleavage (Huai et al., 2017; Palermo et al., 2018; Palermo et al., 2017; Zuo and Liu, 2017). An in silico model (Yoon et al., 2019) that was recently reported also suggested that D861 and N863 are potentially involved in chelating the $Mg^{2+}$ ion at the catalytic center of the HNH domain, although this discovery also remained untested in an experimental setting. In the diverse homologous structures of DNA/RNA nucleases from other species (Yang, 2008; Yang, 2011), however, the residues spatially equivalent to the D861 of SpyCas9 are conserved as an asparagine. The substitution (N62D) of the corresponding asparagine in the active center of bacteriophage T4 Endonuclease VII (T4 Endo VII) has been shown to abrogate its DNA cleavage activity (Biertümpfel et al., 2007). These observations motivated us to examine the potential role of D861 in the HNH domain of Cas9. We mutated D861 to alanine and tested the activity of the D861A variant using an experimental approach based on Cas9-mediated disruption of the *egfp* gene in EGFP-expressing human cells. Unexpectedly, this Cas9 variant exhibited DNA-cleavage activity level similar to that of the wild-type protein (Figure 1e and Figure 1—figure supplement 1). To further validate our finding, we performed in vitro cleavage assays using either plasmid or oligo DNA as a substrate and observed that the D861 variant retained similar activity as the wild type, given enough reaction time (Figure 1—figure supplement 2 and Figure 1—figure supplement 3). Our experiments thus demonstrate that D861 is not critical for HNH domain-catalyzed tDNA cleavage, unlike what would be expected from the reported Cas9 complex structures (Anders et al., 2014; Huai et al., 2017; Jiang et al., 2016; Palermo et al., 2018; Palermo et al., 2017; Zuo and Liu, 2017). In other words, the previously reported structures of the HNH domain of DNA-bound Cas9 (Huai et al., 2017; Palermo et al., 2018; Palermo et al., 2017; Zuo and Liu, 2017) potentially represent a conformation that is incompetent for tDNA cleavage. Hence, we refer to this cleavage-incompetent conformation with an inward-facing D861 as 'psuedoactive state' hereafter. This report aims to unmask the catalytic state of the HNH nuclease domain in Cas9 and explore the underlying mechanism of activation.

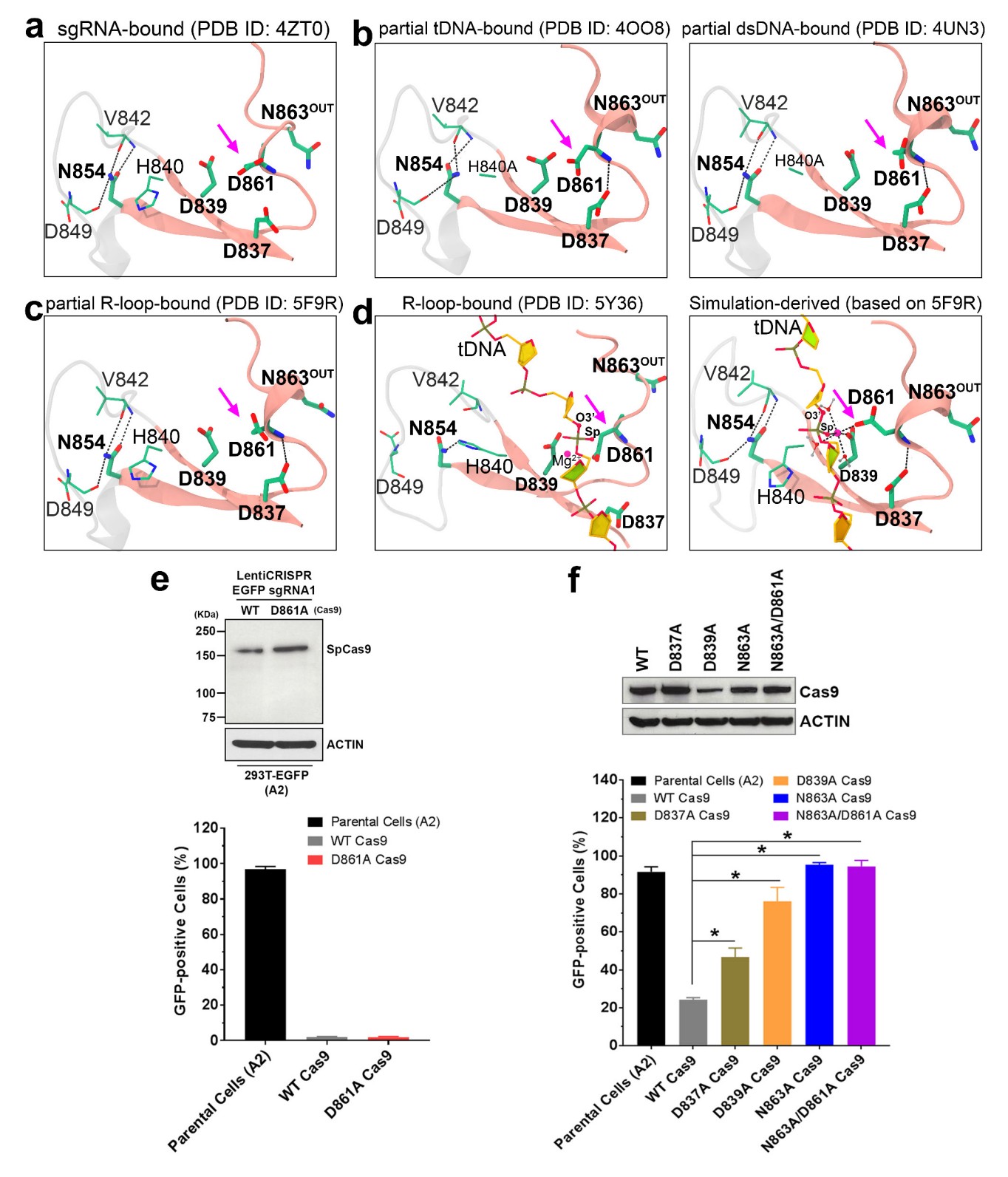

**Figure 1.** Architecture of the HNH domain ββα-Me fold in different binding forms of Cas9 (**a–d**) and site-directed mutagenesis experiments identifying potential catalytic residues (**e–f**). (**a–d**) ββα-Me fold in the sgRNA-bound state of Cas9 (**a**), in the intermediate state (**b**), in the pre-catalytic state (**c**), and in the pseudoactive state (**d**). The ββα-Me fold is represented as pink ribbons, and the residues are shown in stick models and colored by atom type (C, dark green; N, blue; O, red). If present, the bound $Mg^{2+}$ ion is depicted as a magenta sphere, and only the tDNA phosphate-sugar backbone is

*Figure 1 continued on next page*

*Figure 1 continued*

displayed for clarity. The location of the Cas9 D861 is highlighted by an arrow, and the dashed lines denote hydrogen bonds or coordinative bonds. (**e**) The expression and DNA-editing activity of the wild-type and D861A variants of Cas9 paired with an sgRNA sequence that targets the *egfp* gene in HEK293T-EGFP cells. (**f**) The expression and DNA-editing activity of the wild-type and indicated variants of Cas9 paired with an sgRNA sequence that targets the *egfp* gene in HEK293T-EGFP cells. The retention of EGFP expression reflected the loss of activity of Cas9 protein in the cells.

DOI: https://doi.org/10.7554/eLife.46500.003

The following source data and figure supplements are available for figure 1:

**Source data 1.** Numeric data for activity quantification of Cas9 and its variants (XLS).

DOI: https://doi.org/10.7554/eLife.46500.009

**Figure supplement 1.** The DNA sequencing analysis of vectors for expressing different Cas9 variants and the flow cytometry analysis of HEK293T-EGFP cells with different Cas9-sgRNA expression vectors.

DOI: https://doi.org/10.7554/eLife.46500.004

**Figure supplement 2.** Plasmid cleavage activity of SpyCas9$^{WT}$.

DOI: https://doi.org/10.7554/eLife.46500.005

**Figure supplement 3.** Cleavage of radioisotope-labeled oligo DNA substrate by SpyCas9$^{WT}$, SpyCas9$^{D861A}$ and SpyCas9$^{N863A}$.

DOI: https://doi.org/10.7554/eLife.46500.006

**Figure supplement 4.** The architecture of the HNH domain ββα-Me motif in the apo structure of SpyCas9 (**a**) and structural modeling of SpyCas9 with Mg$^{2+}$ ion bound at the catalytic center (**b–c**).

DOI: https://doi.org/10.7554/eLife.46500.007

**Figure supplement 5.** The architecture of the HNH domain ββα-Me motif in the apo structure of *Actinomyces naeslundii* Cas9 (AnaCas9) (**a**) and in that of partial dsDNA-bound *Staphylococcus aureus* Cas9 (SauCas9) (**b**).

DOI: https://doi.org/10.7554/eLife.46500.008

## Results and discussion

Intrigued by the above paradoxical findings between the structural and functional experiments, we performed molecular modeling and molecular dynamics (MD) simulations to further investigate the residues that may participate in the catalysis of tDNA cleavage. We first examined the apo-state crystal structure of Cas9 (*Jinek et al., 2014*). One noticeable feature in this apo-Cas9 structure is that the α-helical element in the HNH domain ββα-metal (ββα-Me) fold appears to pose a unique conformation with N863 pointing toward the catalytic center (*Figure 1—figure supplement 4a*). This inward orientation of N863 was distinct from the inward orientation of D861 observed in the Cas9 structures co-crystalized with sgRNA and/or DNA (*Figure 1a–1d*). However, the N-terminal of the α-helical segment is disordered in this apo-Cas9 structure (*Figure 1—figure supplement 4a*), suggesting a high conformational flexibility around this region. We completed the disordered regions of the apo-Cas9 structure by homology modeling and performed molecular dynamics (MD) simulation with a Mg$^{2+}$ ion placed in the catalytic ββα-Me motif (*Figure 1—figure supplement 4b*). As a result, the Mg$^{2+}$ ion formed an octahedral coordination with D839, N863, and four water molecules (*Figure 1—figure supplement 4c*), which closely resembled the Mg$^{2+}$ coordination in the X-ray crystal structure of *Actinomyces naeslundii* Cas9 (AnaCas9, *Figure 1—figure supplement 5a*) (*Jinek et al., 2014*). Encouraged by this finding, we next applied the above Mg$^{2+}$-bound α-helical conformation in our optimized pseudoactive Cas9 complex structure (*Zuo and Liu, 2017*) (*Figure 2a–2b*) and performed MD-based refinement on the entire structure (see details in Materials and methods). In our final structure, D839, H840 and N863 on the ββα-Me motif formed the catalytic triad that was poised for cleaving the tDNA. The Mg$^{2+}$ ion was coordinated with the residues D839 and N863 of SpyCas9, a scissile phosphate (the pro-Sp oxygen and leaving O3' engaged), and two water molecules in a strained geometry (*Figure 2d–2e*). This structure highly resembled the catalytic configuration present in the X-ray structure of T4 Endo VII/DNA complex (*Biertümpfel et al., 2007*) (*Figure 2c*), indicating the N863 of Cas9 could be engaged in the formation of the catalytic center for tDNA cleavage.

To validate the role of N863 in SpyCas9 function, we examined the gene-editing activities of an N863A variant and a D861A/N863A double-mutant variant using cell-based assays. Both SpyCas9 variants virtually lost the entire gene-editing capability compared to the wild-type protein (*Figure 1f* and *Figure 1—figure supplement 1*). Consistently, our in vitro assays also showed an effective loss of tDNA cleavage activity for the N863A variant, essentially producing only nicked products in the plasmid cleavage assay (*Figure 1—figure supplement 2*) and causing the disappearance of the 23-

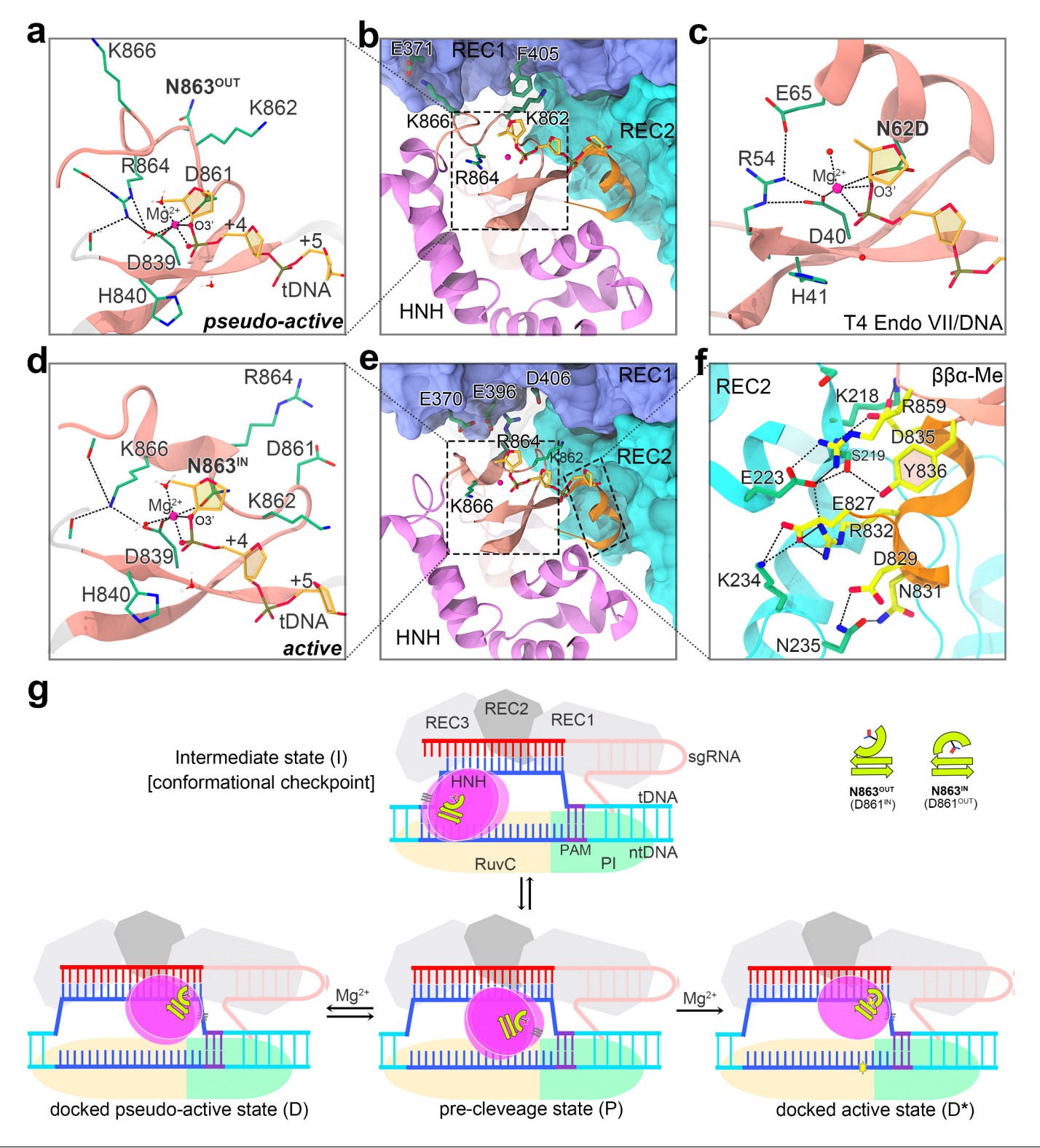

**Figure 2.** Comparison of the active and pseudoactive Cas9-nucleic acid complex structures and proposed mechanism for DNA cleavage activation of Cas9. (a–b) Zoomed-in view (a) and zoomed-out view (b) of the HNH domain docked onto the tDNA and REC lobe in the optimized pseudoactive state. (c) Close-up view of the catalytic configuration of the T4 Endo VII (N62D) ββα-Me motif complexed with a DNA substrate. (d–f) Zoomed-in views (d, f) and zoomed-out view (e) of the HNH domain docked onto the tDNA and REC lobe in the catalytically active state. (g) Schematic diagram of the proposed mechanism underlying Cas9 HNH domain conformational activation.

*Figure 2 continued on next page*

*Figure 2 continued*

DOI: https://doi.org/10.7554/eLife.46500.010

The following source data and figure supplements are available for figure 2:

**Source data 1.** Atomic coordinates of active Cas9-gRNA-DNA complex (in PDB format).
DOI: https://doi.org/10.7554/eLife.46500.018

**Figure supplement 1.** Close-up view of the catalytic centers of some representative ββα-Me superfamily members beyond Cas9.
DOI: https://doi.org/10.7554/eLife.46500.011

**Figure supplement 2.** Multiple sequence alignment of Type II-A and Type II-C Cas9 orthologs focusing on the ββα motif regions.
DOI: https://doi.org/10.7554/eLife.46500.012

**Figure supplement 3.** Lys862 making interactions with Asp837 and/or tDNA as captured in another simulation trajectory of the active state Cas9 complex.
DOI: https://doi.org/10.7554/eLife.46500.013

**Figure supplement 4.** Interactions between the REC2 domain (colored cyan) and the HNH domain formed in the pseudo-active Cas9 complex structure.
DOI: https://doi.org/10.7554/eLife.46500.014

**Figure supplement 5.** Free energy landscapes of the α structure element (residues 859 to 872) in the ββα-Me motif against different sets of reactions coordinates.
DOI: https://doi.org/10.7554/eLife.46500.015

**Figure supplement 6.** The MD-derived metal center configuration in the active (a) and psuedoactive state (b) optimized by DFTB3 QM/MM simulations.
DOI: https://doi.org/10.7554/eLife.46500.016

**Figure supplement 7.** The coordination configuration at the HNH domain active center from our original pseudo-active Cas9 complex structure derived by use of the 12–6 point-charge $Mg^{2+}$ model (viz. the normal usage set in AMBER force field).
DOI: https://doi.org/10.7554/eLife.46500.017

nucleotide tDNA cleavage product in the oligo assay (*Figure 1—figure supplement 3*). We note that the DNA nicks produced in the cell-based assay could be efficiently repaired by the cellular machinery, thereby causing the apparent loss of gene-editing capability in the above N863A variants. Together, our results clearly show N863, instead of D861, is the functional catalytic residue. Some studies have also suggested the possibility of N863 in forming the catalytic center of the Cas9 HNH domain (*Nishimasu et al., 2014*; *Yoon et al., 2019*). In contrast to the observation from another in silico model that was recently published (*Yoon et al., 2019*), our computational model indicated that D861 and N863 are unlikely to be simultaneously involved in the coordination of $Mg^{2+}$ ion for cutting the tDNA. Besides N863, the functional significance of H840 (acting as a general base) has been confirmed by experiments (*Jinek et al., 2012*; *Nishimasu et al., 2014*). Another putative active residue in the HNH domain is D839, which could be inferred from the structures of homologous nucleases (*Yang, 2008*), despite lacking experimental data to support its catalytic role. Here, we discovered that the D839A mutation substantially compromised the gene-editing activity of Cas9 (*Figure 1f*), which is consistent with our simulation studies (*Zuo and Liu, 2017*) and directly supports the significance of D839 for Cas9 activity. Collectively, our structural and functional data corroborate the essence of D839, H840 and N863 for the cleavage function of the HNH domain.

Our MD simulations also excluded the likelihood of D837 and N854 (at the two ends of the β-sheet of the ββα-Me motif) to function as the catalytic residues previously postulated to be directly engaged in the coordination of the $Mg^{2+}$ ion (*Chen and Doudna, 2017b*; *Huai et al., 2017*). The simulations showed that the side chain of N854 is ~10 Å distant from the tDNA, making it unlikely be a residue that directly mediates catalysis. However, an activity assay demonstrated the partial loss of activity in the N854A variant (*Nishimasu et al., 2014*), suggesting that N854 may play an auxiliary role in catalysis. According to our MD models and the solved Cas9 structures (*Figure 1a–1d*), N854 may stabilize the activated conformation of the HNH domain by making several intra-molecular hydrogen bonds within the ββα-Me motif. Moreover, our structural analyses revealed that the Spy-Cas9 N854 is spatially and functionally equivalent to the AnaCas9 N597 and *Staphylococcus aureus* Cas9 (SauCas9) N571 (*Figure 1—figure supplement 5*), indicating this stabilization role may exist in multiple Cas9 orthologs (*Jinek et al., 2014*; *Nishimasu et al., 2015*).

The D837A Cas9 moderately lost gene-editing activity (*Figure 1f*). From the solved Cas9 structures and our simulations, D837 may contribute to the structural integrity of the ββα-Me motif,

although only one hydrogen bond involving its carboxyl group appear to be established (*Figure 1a–1d*). It is also possible that D837 may aid in proper positioning of the tDNA relative to the HNH domain (as described below).

The comparison between the active and pseudoactive Cas9-nucleic acid complex structures revealed distinct interactions that stabilize these two conformational states. In the pseudoactive conformation, R864 appeared to stabilize the catalytic core by forming ionic and hydrogen-bonding interactions within the ββα-Me motif (*Figure 2a*), as observed in various bound forms of Cas9 (*Figure 1a–1d*), whereas K866 is engaged to E371 to form a salt bridge (*Figure 2b*). In contrast, the corresponding residue of R864 is surrogated by K866 in the active state (*Figure 2d*), which is in line with the structures of AnaCas9 (K609) and SauCas9 (K583) (*Figure 1—figure supplement 5*), while R864, on the other hand, is tethered to a negative pocket lined by E370, E396 and D406 on the REC1 domain (*Figure 2e*). Further structural and sequence analysis indicated that the arginine residue, which is at an equivalent position of R864 in the pseudoactive state, is highly conserved across the ββα-Me nucleases beyond Cas9 (*Figure 2—figure supplement 1*), whereas both the Type II-A and II-C Cas9 orthologs feature an invariant lysine residue (*Figure 2—figure supplement 2*) that is spatially equivalent to K866 in the active state. Therefore, a basic amino acid residue (Lys/Arg) seemed to be required at the catalytic core for the ββα-Me superfamily. Moreover, K862 formed aliphatic-aromatic or cation-π interactions with F405 in the pseudoactive state (*Figure 2b*), whereas it appeared to interact with D837 and/or the +5 phosphate (relative to the protospacer adjacent motif) in one of our simulations for the active state (*Figure 2—figure supplement 3*). Despite these differences in the interactions with the REC1 domain, other regions of the HNH domain, especially the helical segment preceding the ββα motif, showed similar patterns of binding to the REC2 domain in the pseudoactive and active states (*Figure 2f* and *Figure 2—figure supplement 4*).

The removal of the interfacial $Mg^{2+}$ ion from our models substantially destabilized the pseudoactive and active conformations of the Cas9-nucleic acid complex by ~600 kcal/mol (*Supplementary file 1*), supporting the indispensable role of the $Mg^{2+}$ ion for Cas9 conformational activation that has been previously demonstrated (*Dagdas et al., 2017*; *Osuka et al., 2018*; *Zuo and Liu, 2017*). The pseudoactive conformation (denoted as N863-OUT hereafter) appeared to be unique to SpyCas9 since the corresponding asparagine residues in other ββα-Me nucleases, regardless of the apo or bound form, typically oriented toward the catalytic center (*Figure 1—figure supplement 5* and *Figure 2—figure supplement 1*). End-state free energy calculations revealed that the N863-OUT SpyCas9 complex is thermodynamically more stable than its catalytically active (N863-IN) form (*Supplementary file 1*). Long time-scale MD simulations based on the isolated HNH domain of Cas9 also showed that the N863-IN conformation is remarkably flexible and tends to transition to the N863-OUT conformation, but not vice versa (*Figure 2—figure supplement 5*). These results, at least partially, explain why the N863-OUT SpyCas9 conformation was captured in the majority of experimental structures (*Anders et al., 2016*; *Anders et al., 2014*; *Hirano et al., 2016*; *Huai et al., 2017*; *Jiang et al., 2015*; *Jiang et al., 2016*; *Jinek et al., 2014*; *Nishimasu et al., 2014*).

To further enhance the confidence in our MD-derived structure models, QM/MM simulations were performed. The metal centers shown in *Figure 2a and d* were optimized by DFTB3 QM/MM simulations (*Figure 2—figure supplement 6*). Compared to our MD simulations, we identified very limited differences from the QM/MD simulations in terms of the active site configuration. Encouragingly, this observation is consistent with our most recent benchmark work (*Zuo and Liu, 2018a*) demonstrating the good performance of the used nonbonded $Mg^{2+}$ ion model (*Liao et al., 2017*) in maintaining challenging enzyme metal centers.

Based on the findings reported here and in the previous studies (*Dagdas et al., 2017*; *Osuka et al., 2018*; *Sternberg et al., 2015*; *Yang et al., 2018*), we present a working model to illustrate how the activation of SpyCas9-mediated DNA cleavage is achieved (*Figure 2g*). The binding of DNA to Cas9-sgRNA induces the interconversion of the HNH domain between the checkpoint intermediate (I) state and the pre-cleavage (P) state. By bridging a gap between the catalytic center and the opposite scissile phosphate, the ambient $Mg^{2+}$ ion subsequently facilitates a conformational change to transition the HNH domain from the P state to the tDNA docked pseudoactive (D, N863-OUT) or active (D*, N863-IN) state. Before the cleavage reaction occurs, both states exist and the conformational population of the two states reaches an equilibrium. The lower-energy pseudoactive state has larger conformational population than the active state. Once the irreversible DNA cleavage

reaction rapidly occurs, the active state conformational population decreases rapidly. The equilibrium between these two states is lost and the conformational population begins to move from the pseudoactive state to the active state until the reaction is completed. The coexistence of multiple Cas9 ternary complexes has important implications for deciphering the biphasic kinetics of Cas9-catalyzed DNA cleavage that has been shown recently by *Raper et al. (2018)*. Taking their results into considerations, we surmise that the early, fast phase characterizes a direct transition from the precatalytic state to the active state, while the slow transition from the pseudoactive to the active state likely contributes to the late, slow phase. Our conclusion would benefit from structural determination of the activated Cas9 ternary complex with metal ions.

Meanwhile, we note that the possibility cannot be ruled out that the N863-IN conformation may be stably formed when enough $Mg^{2+}$ ions are present in the solvent. In other words, the N863-IN conformation could be induced by $Mg^{2+}$ even without engaging a DNA substrate. To test this hypothesis, we have performed additional simulations on the pre-cleavage Cas9 complex (*Jiang et al., 2016*) with multiple $Mg^{2+}$ ions positioned around N863, but were unable to observe the trend toward the N863-IN conformation over µs time scales (data not shown). Consistently, the HNH domain in 4UN3 (*Figure 1b*, right panel) and other crystals (like 5B2R, 5B2T, 5B2S, 5FQ5, 5FW1, 5FW2, 5FW3 and 6AI6) remained in the N863-OUT conformation even though as high as eight $Mg^{2+}$ ions were cocrystalized (*Anders et al., 2016*; *Hirano et al., 2016*; *Nishimasu et al., 2014*; *Olieric et al., 2016*; *Shin et al., 2017*). Therefore, the presence of metal ions alone appears to be insufficient for the HNH domain conformational activation. Nevertheless, further experimental and computational research is needed to elucidate the transition process between the two states shown herein as well as the chemical principles defining the requirement for asparagine instead of aspartic acid.

## Conclusion

Overall, our study has delineated a molecular framework underlying the catalytic conformation of the HNH nuclease domain of SpyCas9. The findings presented here advance our knowledge of conformational activation that enables Cas9-mediated DNA cleavage, set an important foundation for future studies to further understand the structure-function relationships of Cas9, and facilitate the rational design of Cas9 variants in the future.

# Materials and methods

**Key resources table**

| Reagent type (species) or resource | Designation | Source or reference | Identifiers | Additional information |
|---|---|---|---|---|
| Cell line (*H. sapiens*) | HEK-293T | ATCC | CRL-3216 (RRID: CVCL_0063) | N/A |
| Antibody | Cas9 (7A9-3A3) mouse monoclonal Ab | Cell Signaling Technology | 14697 (RRID: AB_2750916) | WB (1:500), nonfat milk blocking |
| Antibody | Anti-ACTIN mouse monoclonal Ab (C4) | MP Biomedicals | SKU08691002 (RRID: AB_2335127) | WB (1:5000), nonfat milk blocking |
| Transfected construct (Synthetic) | pLenti CMV GFP Puro | Addgene | 17448 | *Campeau et al., 2009* Aug 6;4 (8):e6529. |
| Transfected construct (Synthetic) | lentiCRISPR-EGFP sgRNA 1 (WT Cas9) | Addgene | 51760 | Shalem et al, *Science.* 3;343 (6166):84–7. |
| Transfected construct (Synthetic) | lentiCRISPR-EGFP sgRNA 1 (D861A Cas9) | Site-directed mutagenesis (Wang lab) | N/A | N/A |
| Transfected construct (Synthetic) | lentiCRISPR-EGFP sgRNA 1 (D837A Cas9) | Site-directed mutagenesis (Wang lab) | N/A | N/A |

*Continued on next page*

*Continued*

| Reagent type (species) or resource | Designation | Source or reference | Identifiers | Additional information |
|---|---|---|---|---|
| Transfected construct (Synthetic) | lentiCRISPR-EGFP sgRNA 1 (D839A Cas9) | Site-directed mutagenesis (Wang lab) | N/A | N/A |
| Tansfected construct (Synthetic) | lentiCRISPR-EGFP sgRNA 1 (N863A Cas9) | Site-directed mutagenesis (Wang lab) | N/A | N/A |
| Transfected construct (Synthetic) | lentiCRISPR-EGFP sgRNA 1 (N863A/D861A Cas9) | Site-directed mutagenesis (Wang lab) | N/A | N/A |

## Cell culture

Human embryonic kidney 293T (HEK293T) cells (ATCC, Manassas, VA) as a subclone of the HEK293 cell line were cultured in DMEM (Thermo Fisher Scientific, Carlsbad, CA) containing 10% fetal bovine serum (FBS; Thermo Fisher Scientific, Carlsbad, CA) at 37°C. All cells were periodically tested using the MycoAlert mycoplasma detection kit (Lonza, Walkersville, MD) and free of mycoplasma. The HEK293T cells were used to established EGFP-expressing cells by the lentivirus-mediated transduction of pLenti-CMV-GFP-Puro expression plasmid (Addgene, Cambridge, MA) into the cells followed by the selection of single-cell clones that stably express EGPF and fluorescence green (*Campeau et al., 2009*). The stable clone A2 was used in this study.

## CRISPR-Cas9-mediated gene editing

For testing Cas9-mediated editing of the egfp gene in the HEK293T-EGFP (A2) cells, an EGFP-targeting sgRNA sequence (EGFP sgRNA1: 5'GGGCGAGGAGCTGTTCACCG3') was cloned into a lentiCRISPR plasmid (Addgene, Cambridge, MA) and resulted in a construct of a one-vector system for co-expression of sgRNA and wild-type SpyCas9 (Addgene, Cambridge, MA) (*Shalem et al., 2014*). The site-directed mutagenesis was performed to specifically introduce mutations into the *cas9* gene open reading frame (ORF) in the expression construct to generate the expression vectors of different Cas9 variants with the EGFP sgRNA sequence. After mutagenesis, the DNA sequencing of each expression construct was performed to confirm the mutations of the Cas9 gene ORF. HEK293T-EGFP (A2) cells transduced with the Cas9 and sgRNA expression constructs were selected using 5 ug/ml puromycin for two weeks prior to the downstream analysis to determine the editing efficiencies of different Cas9 variants.

## Immunoblotting

The general procedure for immunoblotting was described in previously published reports (*Wang et al., 2008*; *Zolekar et al., 2018*). The primary antibody against SpyCas9 (catalog# ab191468) was obtained from Abcam (Cambridge, MA). HRP-conjugated secondary antibodies were from Jackson ImmunoResearch Laboratories (West Grove, PA).

## Flow cytometry analysis

For determining fluorescence intensity and quantifying the percentages of EGFP fluorescence-positive cells in cell samples, samples (~$5 \times 10^5$ cells per sample) harvested and resuspended in phosphate-buffered saline (PBS) were analyzed using a SH800Z cell sorter (Sony Biotechnology, San Jose, CA).

## Site directed mutagenesis, and protein overexpression and purification

The protein variants, SpyCas9$^{D861A}$ and SpyCas9$^{N863A}$ were produced by sequence independent cloning method (SLIC) using SpyCas9$^{WT}$ template plasmid (Addgene: pMJ806) (*Jinek et al., 2012*) and mutagenic primers (*Supplementary file 2*) (*Scholz et al., 2013*). Sequence confirmed clones were transformed into *Escherichia coli* Rosetta strain 2 (DE3) for protein expression. Overexpression and protein purification were carried out using previously published protocols (*Babu et al., 2019*; *Jinek et al., 2012*).

## sgRNA and substrate DNA preparation

The template for in vitro transcription of *sg*RNA (98-nucleotide long) contained a 20 nt long spacer as previously described (*Babu et al., 2019*; *Nishimasu et al., 2014*). The protocols used for in vitro transcription and sgRNA annealing were as reported previously (*Babu et al., 2019*). For creating target DNA, a 30 nt long protospacer flanked by a PAM (GGG) was introduced into pUC19 (*Babu et al., 2019*).

## Plasmid DNA cleavage assay

The proteins were diluted in 20 mM HEPES pH 7.5, 150 mM KCl, and 2 mM TCEP, and the cleavage assays were carried out in a final volume of 10 µL. The reaction mix contained 20 mM Tris-HCl pH 7.5, 100 mM KCl, 5 mM MgCl$_2$, 5% (v/v) glycerol, 0.5 mM TCEP, 100 ng of substrate plasmid, 50 nM SpyCas9, and 60 nM sgRNA (protein and RNA at a ratio of 1:1.2 molar). The reaction mixture was incubated at 37°C and stopped at different time points (15 s, 30 s, 1 min, 2.5 min, 5 min, 7.5 min, 10 min, 15 min, 30 min, 45 min, and 60 min) by the addition of 50 mM EDTA and 1% SDS. The reaction products were resolved on 1% agarose gel and products were visualized by ethidium bromide staining. The gel was imaged using a BioRad ChemiDoc MP apparatus. To quantify the cleavage activities, each gel image was analyzed using the ImageJ software (*Schneider et al., 2012*). The bands of nicked (N), linear (L), and supercoiled (SC) DNA were quantified and designated as $I_N$, $I_L$, and $I_{SC}$ respectively. The nicked, linear and total activity (TA) was calculated using the following equations:

$$\text{Nicked}(\%) = \left[ \frac{I_N}{I_N + I_L + I_{SC}} \right] \times 100 \qquad (1)$$

$$\text{Linear}(\%) = \left[ \frac{I_L}{I_N + I_L + I_{SC}} \right] \times 100 \qquad (2)$$

$$\text{TA}(\%) = \left[ \frac{I_N + I_L}{I_N + I_L + I_{SC}} \right] \times 100 \qquad (3)$$

For each reported data point, average values were obtained from a minimum of three replications that were performed using proteins produced from two independent preparations to account for variations in active protein fraction between different preparations. Standard deviation (SD) and standard error of mean (SEM) were calculated based on the number of replications using the following equations:

$$SD = \sqrt{\sum (R - R_{AV})^2 / (n-1)} \qquad (4)$$

$$SEM = SD / \sqrt{n} \qquad (5)$$

where $R$ is a data value from each replication, $R_{AV}$ is average of data values of all the replications, and $n$ is the number of replications (a total of three for each protein variant).

## Radioactive assay

Two separate oligo DNA strands used for the radioactive assay were ordered from Integrated DNA technology (IDT, *Supplementary file 2*). Target (T) and non-target (NT) strands were mixed at equimolar concentration in the annealing buffer (30 mM HEPES pH 7.5, 100 mM potassium acetate) and heated at 95°C for 2 min and allowed for slow cooling. The annealed duplex oligo DNA was 5' end labeled with $^{32}$P ($\gamma-^{32}$P ATP purchased from PerkinElmer) using T4 polynucleotide kinase (New England Biolabs). The labeled oligo DNA was purified using BioSpin column P-30 (BioRad). The reaction buffer contained 20 mM Tris-HCl pH 7.5, 100 mM KCl, 10 mM MgCl2, 5% (v/v) glycerol, 0.5 mM TCEP. Approximately 5 nM of labeled oligo duplex was incubated with 250 nM SpyCas9 and 300 nM sgRNA (protein and RNA at a ratio of 1:1.2 molar) at 37°C and stopped at different time points (15 min, 30 min, and 60 min) using EDTA at 10 mM final concentration. Then the samples were treated with proteinase K (New England Biolabs) for 15 min at 50° C to remove SpyCas9. This was followed by addition of equal volume of loading dye (2X concentration is 20 mM EDTA, 95%

formamide, 2% SDS, and 0.025% bromophenol blue). The reaction samples were resolved on a 16% poly-acrylamide gel containing 20% formamide and 6.4 M urea. The bands were visualized by phosphor imaging with Typhoon FLA 7000 system (GE life sciences). Three independent replications were performed using proteins from two independent preparations.

## Molecular modeling and molecular dynamics simulations of apo-Cas9

The initial coordinates of the apo-state SpyCas9 were taken from the Protein Data Bank (PDB) under accession number 4CMP (solvated at 2.6 Å resolution) (*Jinek et al., 2014*). This X-ray structure contains two Cas9 monomers, and the molecule B was considered for modeling here (*Figure 1—figure supplement 4a*). The disordered regions were built up with the tool SWISS-MODEL (*Waterhouse et al., 2018*) and the missing heavy atoms and hydrogens were added by using the *leap* program within AMBERTOOL16 (*Salomon-Ferrer et al., 2013*). The complete structure was then solvated in a cubic water box with a minimal thickness of 13.5 Å from each edge, leading to a periodic boundary box of $138 \times 153 \times 126$ Å$^3$. The system was neutralized by Na$^+$, and additional NaCl was added to generate a physiological ionic strength of 150 mM. The resulting simulation box contains ~230,000 atoms.

The above system was simulated by the CUDA-accelerated version of AMBER16 *pmemd* engine (*pmemd.cuda*; *Salomon-Ferrer et al., 2013*) using the amber force field *ff14SBonlysc* for protein, the TIP3P model for water (*Jorgensen et al., 1983*), and the Joung-Cheatham parameter sets for monovalent ions (*Joung and Cheatham, 2008*). The non-bonded interactions were truncated at 10 Å, and the long-range electrostatics were calculated through the particle mesh Ewald (PME) summation method (*Darden et al., 1993*), with a grid spacing of 1 Å. The bonds involving hydrogens were constrained via the SHAKE algorithm (*Miyamoto and Kollman, 1992*), allowing use of 2-fs time step of integration. After thorough energy minimization, the system underwent slow heating over 50 ps from 0 K to the target 310.15 K in the isothermal-isochoric (NVT) ensemble, followed by a 20-ns equilibration in the isothermal-isobaric (NPT) condition. The protein backbone atoms were restrained in the heating and equilibration stages. Finally, the production run was performed under the NPT ensemble without restraints, extending up to 100 ns. The pressure was controlled at 1.013 bar via the Monte Carlo barostat, and the temperature was maintained at 310.15 K through the Langevin thermostat implemented in AMBER16.

The final structural snapshot from the above simulation was then extracted, and a Mg$^{2+}$ ion was placed at its HNH domain active center to set up the Mg$^{2+}$-bound system (*Figure 1—figure supplement 4*) by reference to the AnaCas9 crystal structure bound with a Mg$^{2+}$ ion (*Figure 1—figure supplement 5a*) (*Jinek et al., 2014*). Also, extra Mg$^{2+}$ ions were introduced into the system to obtain a physiological concentration of 5 mM. The parameter set developed by Li et al. (*Li et al., 2013b*) was selected for Mg$^{2+}$. In the equilibration stage, the distances between the Mg$^{2+}$ and the coordinating oxygens on D839 and N863 was restrained to 2.1 Å (i.e., the experimental ion-oxygen distance; *Zheng et al., 2008*). The production run without restrains was extended to 50 ns.

## Molecular dynamics refinement of the pseudoactive Cas9 complex

The starting structure of the pseudoactive Cas9 complex (*Figure 1d*, right panel) was obtained from our recent work (*Zuo and Liu, 2017*), which was derived by employing the unbiased, brute-force MD simulations on the crystal structure of Cas9-sgRNA-DNA (PDB code: 5F9R) that was captured in the pre-cleavage state (*Jiang et al., 2016*). The structural model has been validated by different experiments, yet the Mg$^{2+}$ ion at the HNH domain catalytic center appeared to lose one critical coordination bond with the leaving group O3' of the scissile phosphate (*Zuo and Liu, 2017*) (*Figure 2—figure supplement 7*), as compared to the homologous T4 Endo VII structure complexed with a DNA junction (*Biertümpfel et al., 2007*) (*Figure 2c*). We reasoned that this issue may be due to the deficiency with the simple point-charge Mg$^{2+}$ model used. Most recently, we systematically evaluated the performance of all four types of non-bonded Mg$^{2+}$ ion models in terms of maintaining a challenging metal center configuration in a nuclease system (*Nowotny et al., 2005*). Our benchmark calculations demonstrated that the multisite models based a 12-6-4 Lennard-Jones potential (*Li and Merz, 2014*; *Liao et al., 2017*), which take charge-induced dipole effects into account, are the only ones that are capable of reproducing the experimental coordination patterns (*Zuo and Liu, 2018a*). Accordingly, the 12-6-4 type multisite model (*Jorgensen et al., 1983*) (here the midC4 set) was

considered for the Cas9 complex simulation, along with the TIP4PEw model for water, the Joung-Cheatham parameter sets for monovalent ions (*Joung and Cheatham, 2008*), and the amber force fields *ff14SBonlysc*, *ff99bsc0_chiOL3*, *ff99bsc0_OL15* for protein, RNA and DNA, respectively. Basically, the complex system was set up and simulated following the above protocol for the apo-Cas9 systems. The generated simulation box is approximately $109 \times 145 \times 166$ Å$^3$ sized, containing ~282,000 atoms. With different random seed numbers, two parallel simulations were carried out by using the latest AMBER18 (*Salomon-Ferrer et al., 2013*) that enables GPU calculations of the 12-6-4 ion potential. The simulation length was set to 200 ns each.

## Molecular modelling and molecular dynamics simulations of the active Cas9 complex

The initial model for the active Cas9 complex was constructed by replacing the α-helical segment of the ββα-Me motif in the optimized pseudoactive Cas9 complex (*Figure 1a*) with the corresponding part in the Mg$^{2+}$-bound apo-Cas9 structure (*Figure 1—figure supplement 4c*). The pseudoactive Cas9 complex structure was taken from the above production simulation near 100 ns (i.e. about half of the simulation time). The Mg$^{2+}$-bound apo-Cas9 structure from the simulation trajectory was selected based on the observation of reasonable bonding with the connecting residues and minimal steric clashes after replacement of the α-helical segment. After sufficient energy minimization, the structural model was subjected to multi-stage equilibration: an initial 20-ns relaxation of the α-helical segment and surrounding residues, an another 20-ns equilibration with the inter-atomic distances within the metal center retrained relative to the T4 Endo VII system (*Biertümpfel et al., 2007*), followed by an additional 20-ns equilibration with the restraints gradually released. Subsequently, two independent replicas were performed (250 ns/run) under the same simulation conditions set for the pseudoactive system above.

## Molecular dynamics simulations of isolated Cas9 HNH domain

Additional MD simulations were performed to investigate the relative stability of the two conformational states (i.e. N863-IN and N863-OUT) of the α structure element containing N863 and D861. The starting coordinates were taken from the respective structure models above, and only the HNH domain of Cas9 (residues 781 to 905) was included in our simulation to enhance the sampling efficiency. Each isolated HNH domain in the two states was immersed in a truncated octahedral water box, with a minimal thickness of 14.5 Å. The ionic centration was set to 100 mM by adding an appropriate number of K$^+$ and Cl$^-$ ions in the aqueous solution. The amber force field *ff14SBonlysc* and the TIP3P model were used for describing the protein and the water molecules, respectively, and the parameter sets for the monovalent ion were derived from the work by *Joung and Cheatham (2008)*. For each system, five independent simulations were performed under the NPT ensemble with different initial velocities, using a timestep of 2-ps. Each replica was extended to ~10 us, yielding a total of ~50 us of sampling for each system.

## Hybrid quantum mechanics/molecular mechanics simulations

The semiempirical DFTB3 QM/MM simulations were further implemented to improve the reliability of our MD-derived structure models. DFTB3 is the third-order variant of density functional theory tight binding (DFTB) that is formulated in a DFT framework (*Gaus et al., 2012*). According to the extensive benchmark calculations, DFTB3 in its current form is most reliably for structural properties, including for fairly complex bimetallic motifs in diverse metalloenzymes (particularly the phosphoryl-transfer enzymes) (*Gaus et al., 2012*; *Lu et al., 2015*; *Lu et al., 2016*; *Roston et al., 2018*). The QM region includes the catalytic Mg2+ ion, the protein residues that coordinate the metal ion (i.e., D839 and D861/N863), the general base H840, part of the scissile phosphate and nearby atoms on the target strand, and the water molecules surrounding the metal ions, H840, and the scissile phosphate (cf. *Figure 2a and d*). Only the side chains of protein and the backbone of DNA are kept in the QM region, and link atoms are added between the Cα and Cβ atoms for the amino acids or between the C4′ and C5′ atoms for the nucleotides. The partitioning results in a total of 72 and 75 QM atoms for the pseudoactive and active Cas9 models, respectively. The dummy complex for the Mg$^{2+}$ ions employed in pure MD simulations is changed back to the realistic single-atom form. The MM part of the protein and nucleic acids are described using the same AMBER force fields as mentioned above,

and the water molecules are described with the TIP3P model. After the stages of energy minimization and slow heating, each system was subjected to two parallel 1,000-ps QM/MM simulations performed with the AMBER program.

## Computation of conformational free energies

The free energies of the Cas9-nucleic acid complex conformers were estimated through the end-point Molecular Mechanics-Generalized Born Surface Area (MM-GBSA) approach (*Miller et al., 2012*). Compared to the alternative Molecular mechanics-Poisson Boltzmann Surface Area (MM-PBSA), MM-GBSA has been proven to be give comparable or even better accuracy in ranking ligand binding affinities as well as in calculating the relative stability of multiple conformations of a biomolecular system, though MM-PBSA is physically more rigorous (*Li et al., 2013a*; *Zuo and Liu, 2016a*; *Zuo et al., 2018b*). For each state, the MM-GBSA calculations were performed over an ensemble of 2000 snapshots extracted from the last 50 ns of the simulation trajectories using the program *MMPBSA.py* in AmberTools16. The pairwise GB model of Hawkins, Cramer, and Truhlar (GB$^{HCT}$) (*Hawkins et al., 1995*; *Hawkins et al., 1996*) was used, with the parameters described by *Tsui and Case (2000)*. The default values of the surface tension and the offset to correct the non-polar contribution to the solvation free energy were adopted and the salt concentration was set to 150 mM. Following our recent works (*Zuo and Liu, 2016a*; *Zuo et al., 2018b*), the two water molecules closest to the Mg$^{2+}$ ion at the HNH domain active center were retained as part of the complex, considering the importance of the interfacial water for binding. The entropic contribution was not taken into account due to high computational demand and potential convergence problem, yet omission of this term does not qualitatively affect the results as previously suggested (*Hou et al., 2011*; *Li et al., 2013a*; *Zuo et al., 2018b*; *Zuo et al., 2016b*).

## Acknowledgements

We appreciate the assistance from Dr. Qinghua Liao (Uppsala University) in preparing the Mg$^{2+}$ ion model for MD simulations and thank the Texas Advanced Computing Center (TACC) at the University of Texas at Austin and the University of North Texas (UNT) High Performance Computing Services (a division of the University Information Technology with additional support from UNT Office of Research and Economic Development) for providing us with computational powers that our study has leveraged.

## Additional information

### Funding

| Funder | Grant reference number | Author |
|---|---|---|
| Shanghai Municipal Education Commission | Program for Professor of Special Appointment at Shanghai Institutions of Higher Learning | Zhicheng Zuo |
| National Science Foundation | MCB-1716423 | Rakhi Rajan |
| National Institute of General Medical Sciences | P20GM103640 | Rakhi Rajan |
| University of North Texas (UNT) Health Science Center | Start-up Fund and Faculty Pilot Grant | Yu-Chieh Wang |
| University of North Texas (UNT) Health Science Center | Start-up Fund and Basic Research Seed Grant | Jin Liu |

The funders had no role in study design, data collection and interpretation, or the decision to submit the work for publication.

### Author contributions

Zhicheng Zuo, Conceptualization, Data curation, Formal analysis, Validation, Investigation, Visualization, Writing—original draft, Writing—review and editing; Ashwini Zolekar, Victor JT Lin,

Formal analysis, Validation, Investigation; Kesavan Babu, Formal analysis, Validation, Investigation, Writing—review and editing; Hamed S Hayatshahi, Formal analysis; Rakhi Rajan, Formal analysis, Validation, Investigation, Visualization, Writing—review and editing; Yu-Chieh Wang, Conceptualization, Formal analysis, Validation, Investigation, Visualization, Writing—original draft, Writing—review and editing; Jin Liu, Conceptualization, Formal analysis, Supervision, Validation, Investigation, Visualization, Writing—original draft, Project administration, Writing—review and editing

#### Author ORCIDs

Zhicheng Zuo (iD) https://orcid.org/0000-0003-2761-5841
Hamed S Hayatshahi (iD) https://orcid.org/0000-0001-8639-7130
Rakhi Rajan (iD) https://orcid.org/0000-0002-8719-4412
Yu-Chieh Wang (iD) https://orcid.org/0000-0002-7445-4561
Jin Liu (iD) https://orcid.org/0000-0002-1067-4063

#### Decision letter and Author response

Decision letter https://doi.org/10.7554/eLife.46500.023
Author response https://doi.org/10.7554/eLife.46500.024

## Additional files

#### Supplementary files

• Supplementary file 1. Free energies of Cas9 ternary complexes.
DOI: https://doi.org/10.7554/eLife.46500.019

• Supplementary file 2. Primers used to construct Cas9 variants.
DOI: https://doi.org/10.7554/eLife.46500.020

• Transparent reporting form
DOI: https://doi.org/10.7554/eLife.46500.021

#### Data availability

The data generated or analysed during this study are included in the manuscript and supporting files. Source data files have been provided for Figures 1 and 2.

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
