## [Decision Letter]

Thank you for submitting your article "Structural and functional insights into the bona fide catalytic state of the CRISPR-Cas9 HNH nuclease domain" for consideration by *eLife*. Your article has been reviewed by three peer reviewers, and the evaluation has been overseen by a Reviewing Editor and Cynthia Wolberger as the Senior Editor. The reviewers have opted to remain anonymous.

The reviewers have discussed the reviews with one another and the Reviewing Editor has drafted this decision to help you prepare a revised submission.

Summary:

Despite considerable effort, there is still no consensus about which residues are necessary for Cas9-mediated cleavage of DNA. In this focused study, the authors combine structural modeling and MD simulations to identify putative residues (i.e., D839-H840-N863) in the HNH domain of Cas9 that are necessary for cleavage of the complementary DNA. The importance of these residues, as well as previously implicated residues, is tested using an in vivo reporter assay. While the reviewers agree that this study helps clarify the catalytic mechanism, the experiments do not provide direct evidence that D839, H840 and N863 are necessary for DNA cleavage.

Essential revisions:

The reviewers raise a few concerns that must be adequately addressed before the paper can be accepted. Some of the required revisions will require further experimentation.

1) Use direct methods to measure the efficiency of DNA cleavage for WT and mutant Cas9s. In vitro biochemistry performed using the preferred enzymes is preferred.

2) MM-GBSA calculations are not accurate. The reviewers suggest more rigorous methods (e.g. umbrella sampling, metadynamics, etc.) to obtain the free energy landscape between the two different conformational states of the loop containing N863 and D861.

---

## [Author Response]

Essential revisions:The reviewers raise a few concerns that must be adequately addressed before the paper can be accepted. Some of the required revisions will require further experimentation.1) Use direct methods to measure the efficiency of DNA cleavage for WT and mutant Cas9s. In vitro biochemistry performed using the preferred enzymes is preferred.

To fully address this concern, we set up a new collaboration with Dr. Rakhi Rajan. Rajan lab performed in vitro plasmid cleavage assays using the two variants of SpyCas9 (viz. SpyCas9^D861A^ and SpyCas9^N863A^), with the wild-type enzyme (SpyCas9^WT^) as a control. The results are shown in Figure 1—figure supplement 2. SpyCas9^WT^ cleaved >95% of the plasmid within 15 seconds and ~80% linearization was obtained within 2.5 mins (Figure 1—figure supplement 2A). The activity profile of SpyCas9^D861A^ is similar to that of SpyCas9^WT^, although SpyCas9^D861A^ appeared to require slightly longer time to linearize 80% of the same plasmid. Based on our results, SpyCas9^D861A^ cleaved >95% of the plasmid in 5 mins, with ~80% linearization within 30 mins (Figure 1—figure supplement 2B). Interestingly, SpyCas9^N863A^ produced only nicked products even after 60-minitue incubation (Figure 1—figure supplement 2C). Time required for SpyCas9^N863A^ to nick ~80% of the plasmid was estimated around 10 mins. The in vitro cleavage assays here strongly support the conclusion from our cell-based experiments and in silico simulations that N863, instead of D861, is the essential for the catalytic reaction performed by the HNH domain of SpyCas9.

Furthermore, we performed radioactive assays using duplex oligos labeled with ^32^P at 5’ end of both strands to determine which endonuclease domain is inactive. SpyCas9^WT^ predominantly produced one band for target strand cleavage by HNH (23 nt long) and one band for non-target strand cleavage by RuvC (31 nt long) (Figure 1—figure supplement 3). SpyCas9^D861A^ also cleaved both target and non-target strands, although the rate of cleavage is relatively lower compared to SpyCas9^WT^. In contrast, there was no product formation from target strand cleavage by SpyCas9^N863A^, clearly indicating that the HNH domain is impaired in this variant. Interestingly, the SpyCas9^N863A^ has significantly reduced total activity, as indicated by the amount of uncut duplex present in the reaction even after one-hour incubation (Figure 1—figure supplement 3).

Altogether, our results indicate that N863 is indispensable for the HNH nuclease activity, whereas D861 appears to provide a supporting role to enable a faster reaction rate but is nonessential for target DNA strand cleavage. A closer examination of our simulation trajectories led us to propose that D861 might have a role in stabilizing the catalytic ββα-Me motif by forming an intra-molecular salt bridge (Figure 2D and Figure 2—figure supplement 6A), or aid the initial recruitment of metal ions around the active center along with other negative species like D839 and D837 (Figure 2A and Figure 2—figure supplement 6B) (Picot et al., 2017, Nat. Commun.). The in vitro experimental results are illustrated in Figure 1—figure supplement 2 and Figure 1—figure supplement 3, and other relevant information have been included in our revised manuscript.

2) MM-GBSA calculations are not accurate. The reviewers suggest more rigorous methods (e.g. umbrella sampling, metadynamics, etc.) to obtain the free energy landscape between the two different conformational states of the loop containing N863 and D861.

We agree that the end-state MM-GBSA approach is physically not so rigorous as umbrella sampling (US) and metadynamics in estimating free energy and should be regarded as semiquantitative in nature. In this study, the difference in free energy between the two conformational states (denoted N863-IN and N863-OUT) is of our most interest, and MM-GBSA is especially advantageous in treating such kinds of calculations due to its good balance of computational efficiency and accuracy as well as due to the absence of the need of reaction coordinate definition (Hou et al., 2011; Miller et al., 2012). As demonstrated by extensive benchmark calculations (Hou et al., 2011), the accuracy of MM-GBSA is generally desirable for problems where trends are of special interest, despite being system-dependent in some cases. Specifically, with this method, we and other groups have obtained reasonable results in (relative) free energy calculations on the same Cas9 system and other homologous systems in recent studies (Zuo and Liu, 2016; Zuo and Liu, 2017; Zheng, Shi and Mu, 2018; Zuo and Liu, 2018).

We fully agree with the reviewer that determining the free energy profile between the two conformational states would benefit a deeper mechanistic understanding of the conformational activation in CRISPR-Cas9. Concerning the case here, however, it is quite challenging to construct the expected PMF profiles with an acceptable accuracy in two months, the given revision time. The loop of ~14 amino acids (denoted α structure element in the text due to historical reason) undergoes a substantial conformational reorganization in between the N863-OUT and N863-IN state (RMSD of ~6 Å). It is nontrivial to define an appropriate collective variable (CV) to describe this process. More importantly, the conformational activation of the HNH domain is aided by concerted interactions with the metal ions and the target DNA, and it is almost infeasible to characterize the complex and subtle interactions properly with a single CV or even a set of CVs. Moreover, the systems here are quite large (containing ~230,000 atoms), and the computational burden needed to derive a converged, multi-dimensional PMF (assuming ideal CVs are available) would be quite massive. Therefore, we think this open topic might warrant a separate, more focused research in the future (as we had noted in Discussion of the manuscript).

Despite the difficulties, we have conducted a set of five unconstrained, long-time scale MD simulations starting from each conformational state using the isolated HNH domain (see details for the simulation protocols in Materials and methods). The resultant free energy landscapes shown in Figure 2—figure supplement 5A-5B were constructed based on the principal component analysis over 50-μs of sampling and are plotted against the first two principalcomponents (PC1 and PC2). We also calculated the free energy landscape against the RMSD of the α structure element and the difference in distances of D839-D861 (d_D839-D861_) and D839-N863 (d_D839-N863_) using the combined simulation trajectories (see details in the legend).

The N863-OUT energy landscape is characterized by a single minimum around the simulation initial point (Figure 2—figure supplement 5B), indicating the N863-OUT conformation is quite stable. By contrast, the N863-IN system is remarkably flexible as shown by the accessibility of much broader conformational space (Figure 2—figure supplement 5A). The observation is well consistent with the X-ray crystallographic data, adding another support to our finding presented in this study.

The entire transition between the N863-OUT and N861 conformations have not been captured in our simulations. Yet we observed that the N863-IN conformation appears to trend toward the N863-OUT conformation via a metastable and a prominent intermediate state (Figure 2—figure supplement 5C). The metastable state is characterized by hydrogen bond interactions between N863 and D861 in the α structure element, with N863 being at a position between the IN and OUT conformation. In contrast, the α structure element in the intermediate state is largely extended, where N863 faces outward and N861 orients partially inward. In the meanwhile, we note that care should be exercised in interpreting the current free energy results, as the use of two CVs only still has difficulty in accounting for such a complex transition process. Overall, the free energy calculations qualitatively support our main conclusion from this work.